# New Generation of Compositional Aquivion^®^-Type Membranes with Nanodiamonds for Hydrogen Fuel Cells: Design and Performance

**DOI:** 10.3390/membranes12090827

**Published:** 2022-08-24

**Authors:** Oleg N. Primachenko, Yuri V. Kulvelis, Alexei S. Odinokov, Nadezhda V. Glebova, Anna O. Krasnova, Lev A. Antokolskiy, Andrey A. Nechitailov, Alexander V. Shvidchenko, Iosif V. Gofman, Elena A. Marinenko, Natalia P. Yevlampieva, Vasily T. Lebedev, Alexander I. Kuklin

**Affiliations:** 1Institute of Macromolecular Compounds, Russian Academy of Sciences, 199004 St. Petersburg, Russia; 2Petersburg Nuclear Physics Institute Named by B. P. Konstantinov of National Research Center “Kurchatov Institute”, 188300 Gatchina, Russia; 3Russian Research Center of Applied Chemistry, 193232 St. Petersburg, Russia; 4Ioffe Institute, 194021 St. Petersburg, Russia; 5Physics Faculty, St. Petersburg State University, 198504 St. Petersburg, Russia; 6Frank Laboratory of Neutron Physics, Joint Institute for Nuclear Research, 141980 Dubna, Russia

**Keywords:** proton exchange membrane, ion-conducting, Aquivion, nanodiamonds, small-angle neutron scattering, membrane–electrode assembly, fuel cell, high temperature operation, stabilizing effect

## Abstract

Compositional proton-conducting membranes based on perfluorinated Aquivion^®^-type copolymers modified by detonation nanodiamonds (DND) with positively charged surfaces were prepared to improve the performance of hydrogen fuel cells. Small-angle neutron scattering (SANS) experiments demonstrated the fine structure in such membranes filled with DND (0–5 wt.%), where the conducting channels typical for Aquivion^®^ membranes are mostly preserved while DND particles (4–5 nm in size) decorated the polymer domains on a submicron scale, according to scanning electron microscopy (SEM) data. With the increase in DND content (0, 0.5, and 2.6 wt.%) the thermogravimetric analysis, potentiometry, potentiodynamic, and potentiotatic curves showed a stabilizing effect of the DNDs on the operational characteristics of the membranes. Membrane–electrode assemblies (MEA), working in the O_2_/H_2_ system with the membranes of different compositions, demonstrated improved functional properties of the modified membranes, such as larger operational stability, lower proton resistance, and higher current densities at elevated temperatures in the extended temperature range (22–120 °C) compared to pure membranes without additives.

## 1. Introduction

The global task of the 21st Century has become protection of the environment, which is predetermined the development of new technologies in the field of energy conversion. The problem of decarbonization throughout the economic and energy development of the world requires new approaches to obtain “clean” energy. In 2015, the Paris Agreement on the climate conservation was adopted, which obliges the states to halve CO_2_ emissions by 2040 with the transition to the use of carbon-free energy carriers, primarily hydrogen [1,2,3,4].

One of the most important areas in the decarbonization of economics, along with solar, wind, and hydropower, is the development of hydrogen energy. Hydrogen energy is based mainly on membrane technologies that use various types of membranes in the composition of fuel cells (FC), both polymeric and inorganic, operating in a wide temperature range, from −30 to 1000 °C, with the possibility of using hydrogen, syngas, hydrocarbons, alcohols, etc. [5,6,7,8,9]. The FC industry is the key technology that justifies the transition to hydrogen energy from energetical and economical points of view. At the same time, the development of hydrogen energy can be combined with the use of renewable energy. Hydrogen technologies and fuel cells can be effectively used in combination with renewable energy sources (solar and wind energy) to increase the stability of energy generation using renewable sources [10].

One of the most dynamically types of FCs that has been developing in recent years are PEMFCs—polymer electrolyte membrane fuel cells or proton exchange membrane fuel cells. The modern state of PEMFC technology is based mainly on perfluorosulfonic acid membranes, which have high physical and mechanical properties, chemical resistance, and high ionic (proton) conductivity. Due to the perfluorinated chemical structure of the PFSA membrane backbone, which is stable against peroxide attack, a high operational durability of such membranes, and of composites based on them, during the operation of a hydrogen fuel cell is achieved, reaching tens of thousands of hours [5,11]. The main type of perfluorinated membranes used in PEMFCs are Nafion^®^ membranes and their analogues [9,12,13,14,15,16], which are based on copolymers of tetrafluoroethylene with perfluorinated sulfonated monomers of various lengths. In recent years, research has been dynamically developing in the field of using perfluorinated Aquivion^®^ membranes with short side chains, which allow the ability to maintain high proton conductivity at temperatures of up to 130 °C, in contrast to long-side-chain Nafion^®^ membranes [17,18,19,20,21], which has advantages for the use of Aquivion^®^ membranes in PEMFCs.

PEMFCs are widely used in vehicles and stationary, micro-combined heat and power (micro-CHP) plants [22,23,24]. The range of applications for PEMFCs is wide, from direct-fired FC units running on pure hydrogen in vehicles to micro-CHP plants running on reformed natural gas or liquefied petroleum gas. According to estimates [25], the power provided by PEMFCs during the decade 2012–2021 increased by almost 30 times, from 68.3 MW to 1998.3 MW (Figure 1), while the PEMFC device power fraction among all types of FC devices increased from 40 to ~85%.

Presently, the task of many researchers improving the characteristics of PEMs for use in various types of fuel cells has become the optimization of their basic properties by introducing various modifiers to enhance the physical–mechanical, electrochemical, and diffusion properties of composite membranes, as well as their thermal stability and ability to maintain the required moisture content in the membrane materials at elevated temperatures [26,27,28,29,30,31,32]. The development of polymer composites with organic and inorganic nanosized functionalized fillers requires the targeted formation of the structure of materials by optimally choosing the nanosized particles. Such composites using functionalized nanofillers should improve the functional characteristics of membrane materials—proton conductivity, water retention at elevated temperatures, mechanical strength, and their performance in membrane electrode assemblies (MEA). Thus, opportunities have opened up for the creation of new materials with unique properties that the original polymer material could not possess [33,34,35].

Among the promising nanosized fillers for composite PEMs, we can highlight detonation nanodiamonds (DND), which are carbon nanoparticles with a diamond crystal structure. A DND, due to the chemical inertness of its core and the possibility of its surface functionalization, makes it possible to obtain particles with acidic, hydroxyl, lactone, and other functional groups on the surface. In recent years, the problem of obtaining diamond single crystals without defective, graphite-like layers (sp^2^ hybridization) around the diamond core has been solved [36]. This made it possible to obtain DND particles of 4–5 nm in size, without graphitized shells, and to control the sign of the charge on the nanodiamond facets with their functionalization. The preparation of the stable hydrosols of functionalized DNDs containing surface carboxyl, hydroxyl, and lactone functional groups with a negative charge sign on the surface (DND Z-) is described in [37]. Using such DNDs, composites based on PEMs of the Aquivion^®^-type were obtained [38,39,40] in which the DND particles were not distributed uniformly in the polymer matrix, but rather, they were assembled into large clusters at the boundaries of the hydrophobic and hydrophilic zones of the copolymer. Such clusters work as proton accumulators that increase the proton conductivity at a low DND content (0.5 wt.%). However, a higher loading of a polymer matrix with DND Z- leads to the growth of inhomogeneities distribution and conductivity lowering. Composite membranes based on Aquivion^®^ with nanodiamonds were also obtained in [41], where a strong increase in the proton conductivity of the membranes was achieved, but only at low relative humidity (RH).

DNDs with sulfonic acid functional groups are also interesting for the loading of PEMs. Their preparation and use in the composites of the Aquivion^®^-type is described in [42]. The profit in the use of sulfonated DNDs is due to their ability to supply additional ionogenic groups to the membrane and stimulate a formation of the conductive channels in the polymer matrix to increase the proton conductivity of such composites by enforcing their water-retaining properties at high operating temperatures. This is important for the operation of the composites in an MEA. However, the sulfonated derivatives of DNDs are very difficult to synthesize [42].

A DND with a positively charged surface (DND Z+, protonated) [43] seems to be more promising for modifying the conductivity mechanism in membranes. In such composites, a larger gain in the proton conductivity is achieved due, presumably, to a greater degree of positive DND integration into the polymer matrix, with negative charges on sulfonic acid groups [39]. Based on the experience of our studies on the use of DNDs with various functional groups in composites with perfluorinated sulfonic acid copolymers, all of the electrochemical tests were performed on the composites of Aquivion^®^-type copolymers with DND Z+.

Functionalized DNDs can create conduction channels in the polymer matrix of conducting copolymers due to the possibility of their association into linear (branched) fractal structures [38,39]. In such membranes, the intense migration of protons along the surface of diamond particles by the hopping mechanism through the proton adsorption centers and sulfonic acid groups of the side chains of the perfluorinated copolymer should be expected, similar to [44,45]. This mechanism can be implemented by the functionalized surfaces of nanodiamond particles, which act as proton accumulators when a significant free charge is created at the diamond–polymer interface. The proton transport through the membranes can be organized in two stages: first, the protons are accumulated on the surface of the diamonds, then, due to the charge gradient that has arisen, the protons overcome the activation diffusion barriers and migrate along the diamond particles bound in a chain along the diamond–polymer interface. In this case, the regular diffusion channels will work, rather than the cavities that arise when small, free volume elements are randomly bound between the polymer chains, as in conventional membranes (Nafion^®^ or Aqiuvion^®^) [40]. The active diamond component is able not only to form the regular stable conduction channels, but also to structure the polymer matrix in general, increasing its mechanical strength while maintaining the conductive functional properties of the composite membranes. At the same time, the common conduction channels are also formed in such composites, and the transport of protons according to the traditional Grotthuss scheme for perfluorinated sulfonic acid membranes will be preserved.

The aim of this work is to find the relationship between the structure of the composite membranes, based on the Aqiuvion^®^ type, and perfluorinated copolymers with protonated DNDs, as well as their properties, and to study the effect of DND additives on the electrochemical properties of the composite membranes, their water-retaining capacity, and their performance in membrane–electrode assemblies at elevated temperatures.

## 2. Experimental

### 2.1. Samples Preparation

#### 2.1.1. Preparation of Aquivion^®^-Type Membranes with and without Nanodiamonds

Perfluorinated membrane copolymers of the Aquivion^®^-type, obtained by aqueous-emulsion technology, were used as the matrix material [46,47]. The technological process of obtaining the copolymer precursors of the Aquivion^®^-type membranes in the mode of an aqueous-emulsion copolymerization of tetrafluoroethylene (TFE) with perfluorinated monomer perfluoro-3-oxapentensulfonyl fluoride is considered in detail in our works [21,48].

To prepare composites of the Aquivion^®^-type with DNDs, two copolymers with close equivalent weights (EW) of 890 and 897 g-eq/mol (average EW 894 ± 7 g-eq/mol) were used (analogues of the Aquivion^®^ E-87 brand from Solvay Specialty Polymers USA, LLC (Alpharetta, GA, USA)).

The membranes were prepared by casting from solutions of a short-side-chain copolymer in SO_3_Li form in dimethylformamide (DMF) with concentrations of ~2% wt. on the polished glass surface. The solvent was slowly removed during the heating of the surface, followed by annealing. The resulting films were washed out with 15% nitric acid to transform the membranes into an SO_3_H form. The process of preparing the membranes was described in detail in our previous work [49]. The compositional membranes were cast from the mixture of the copolymer with DNDs in a DMF medium, as we described in our previous works [38,39,40]. To prepare such mixtures, the colloidal solutions of DNDs in a DMF were previously prepared, which required special treatment of the initial aqueous dispersions of the DNDs, depending on the charge sign of their surfaces. In the case of positively charged nanodiamonds, DND Z+, their initial hydrosol was dried, the powder was dispersed in a DMF using powerful ultrasonication treatment, and the resulting suspension of DND Z+ in the DMF was additionally centrifuged to separate the large aggregates [39].

Thus, the composites based on the Aquivion^®^-type membrane with a DND Z+ content in the range of 0.25–5.0 wt.% were obtained in the form of solid films with a thickness of 40–70 μm.

#### 2.1.2. MEA Preparation

The following materials were used to prepare the membrane–electrode assemblies (MEA):(1)Aquivion^®^-type membranes in an SO_3_H^+^ form with an EW of 897 ± 3 g-eq/mol and different DND contents (0, 0.5, and 2.6% wt.) were used as the proton-conducting membrane;(2)Platinized carbon black (Pt/C) containing 40% Pt, a commercial product of the E-TEK brand, was used as an electrocatalyst;(3)A perfluorinated short-side-chain (Aquivion^®^-type) ionomer solution in an SO_3_H^+^ form with 4.2 wt.% and an EW of 790 g-eq/mol in an n-propanol/water/ethanol mixture;(4)N-propanol; and(5)Deionized water (R > 18 MΩ cm).

Preparation of the catalyst Inks

The technological operations for preparing the dispersion of an electrode material included two stages: the mechanical and ultrasonic dispersions of the mixture of accurate samples of components in the n-propanol–water mixture. The volume ratio of n-propanol to water liquid components was 1:1. The ratio of the solid to liquid phases in the final dispersion in this case was 1:30.

Mechanical dispersion was performed in a Milaform MM-5M magnetic stirrer (Milaform-service, Neftekamsk, Russia) with a velocity of core rotation of approximately 400 rpm (the core was placed into a plastic case) to obtain a visually homogeneous mixture (without visible blobs) for ~0.5 h. The subsequent ultrasonic dispersion was carried out in a Branson 3510 ultrasonic bath (Branson Ultrasonics Corp., Danbury, CT, USA) for 40 h to obtain a homogeneous dispersion that did not experience separation after 1 min.

The MEA was produced by applying a fine dispersion of components in a mixture of n-propanol–water (catalyst inks) onto the surface of a proton-conducting membrane consecutively on both sides. The membrane was preliminarily thermostated at 85 ± 5 °C and the area of application of the electrode material (1 × 1 cm^2^ in size) was limited by a stainless-steel mask. The loading of components was controlled gravimetrically. As a result, two samples of composite membranes with different fractions of DNDs were prepared. The size of the electrodes was 1 × 1 cm, and they were located in the center on both sides of the membrane, which was 5 × 5 cm^2^ in size. The loading of platinum was 0.40 ± 0.06 mg/cm^2^.

### 2.2. Methods of Characterization

#### 2.2.1. Proton Conductivity Measurements

The proton conductivity was measured by an impedance spectroscopy at 20 and 50 °C in a state of equilibrium saturation with water. The maximum water content in the membrane was achieved by boiling it at 100 °C for 1 h, which is equivalent to the conditions of ultimate moisture content (RH = 100%). A Z-3000X impedance meter (Elins, Moscow, Russia) with the four-electrode scheme of measuring cell connection was used with a range of frequency of 10–150,000 Hz. The proton conductivity (σ_H_) was calculated by Equation (1):σ_H_ = *L*/(*R* · *h* · *b*)(1)
where σ_H_ is the specific conductivity of the membrane (S·cm^–1^), *L* is the distance between the voltage electrodes of the measuring cell (cm), *R* is the protonic resistance of the membrane (Ohm), *h* is the average thickness of the membrane (cm), and *b* is the average width of the membrane (cm).

#### 2.2.2. Scanning Electron Microscopy

The surface structure of the obtained membranes was studied by scanning electron microscopy (SEM). Micrographs were obtained using a Zeiss AURIGA Laser (Carl Zeiss, Jena, Germany) multifunctional analytical system with crossed ion and electron beams. The system was equipped with a GEMINI^®^ electron optical column (Carl Zeiss, Jena, Germany), and a field emission cathode was used as an electron source. An In-Lens and an Everhart-Thornley SE2 (Carl Zeiss, Jena, Germany) were used as secondary electron detectors to obtain the images. The system parameters in the SEM mode were: magnification of 12 × −1,000,000 ×, spatial resolution of 1 nm at an accelerating voltage of 15 kV, beam current of 10 pA–20 nA, and accelerating voltage in the range of 0.1–30 kV. The resulting images were processed using the SmartSEM^®^ 6.00 software package (Carl Zeiss, Jena, Germany).

#### 2.2.3. Stress–Strain Mechanical Tests

Mechanical tests on the membranes were carried out using the uniaxial extension mode of an AG-100X Plus universal setup for mechanical tests (Shimadzu Corp, Kyoto, Japan). The working length of the samples was 25 mm and the extension velocity was 100 mm·min^–1^. The tests were carried out under the following controlled climatic conditions: relative humidity (RH) in the air of 50 ± 5% and a temperature of 23 ± 1 °C. The following characteristics of the material were determined in the tests: elastic modulus E, yield strength σ_Y_, ultimate tensile strength σ_T_, and ultimate deformation before destruction ε_D_.

#### 2.2.4. Small-Angle Neutron Scattering

The experiments on small-angle neutron scattering (SANS) were performed with the YuMO instrument (Joint Institute for Nuclear Research, Dubna, Russia) in the range of a momentum transfer of *q* = (4π/λ)sin(θ/2) = 0.06 ÷ 6 nm^−1^, where θ is the scattering angle and λ = 0.5–8 Å—neutron wavelength [50]. This range of *q* made it possible to reveal structural features on a scale of ~2π/*q* ~ 1–100 nm. The scattering cross-sections dΣ/dΩ vs. *q* were evaluated considering the recalculation for the thickness of the samples and for the measurements of vanadium as the standard of the known scattering cross section using the SAS software package [51]. The membranes, packed in several layer stacks and covered with aluminum foil transparent for neutrons, were tested in an air-dry state at ambient conditions.

To process the SANS data for the pure membrane without diamonds, we used the previously developed structural model, Equation (2), characterizing the scattering on cylindrical objects (channels) having the transversal radius of gyration *R_g_* and specially organized in a polymer matrix. In Equation (2), the fractal exponent *n* characterizes the geometry of the channels (straight, curved, or branched), and a mutual arrangement of the channels is described by pair correlations at several characteristic distances (*R_i_*) [52].
(2)$$\frac{d\Sigma }{d\Omega }\left(q\right)={\frac{I_{0}R_{g}^{3}}{{\left(qR_{g}\right)}^{n}}e}^{-\frac{{\left(qR_{g}\right)}^{2}}{2}}\left(1+\overset{k}{\underset{i=1}{\sum }}C_{i}\frac{\sin qR_{i}}{qR_{i}}\right)+B$$

The square of the form factor in the model ${\frac{1}{{\left(qR_{g}\right)}^{n}}e}^{-{\left(qR_{g}\right)}^{2}/2}$ generally describes curved, as well as branched, cylindrical channels (form factor of a cylinder ${\frac{1}{q}e}^{-{\left(qR_{g}\right)}^{2}/2}$) having a transversal radius of gyration *R_g_* (the channel diameter is 2$\sqrt{2}$*R_g_*) when the transverse dimension of the channel is much less than its length (*R_g_* << *L*). This asymptotic approximation is valid in the range of momentum transfer 1/*L* << *q* ≤ 1/*R_g_* for the form factor of the structures where their geometry is determined by the fractal exponent *n* [53,54]. The value *n* = 1 corresponds to straight cylinders, 1 < *n* < 2 corresponds to curved cylinders, *n* = 2 characterizes a statistically curved channel similar to a Gaussian polymer chain, and 2 < *n* < 3 corresponds to a branched structure. The structure factor $1+\sum_{i=1}^{k}C_{i}\frac{\sin qR_{i}}{qR_{i}}$ describes pair spatial correlations of the channels at several (*k* = 4…6) characteristic distances (*R_i_*). *C_i_* is the average number of objects correlating with the selected object at a distance *R_i_*. The parameter *I*_0_ characterizes the intensity of forward scattering, depending on the scattering ability of the structural elements. *B* is the constant (incoherent background).

#### 2.2.5. Thermogravimetric Analysis

Thermogravimetric analysis of the membrane samples in an air-dried condition was carried out on a Mettler-Toledo TGA/DSC 1 derivatograph with STAR^e^ System software 16.00 (Mettler-Toledo GmbH, Greifensee, Switzerland), with air blowing through the derivatograph chamber at a flow rate of 30 cm^3^∙min^−1^ in the mode of a uniform temperature rise at a rate of 10 K∙min^−1^ in the temperature range of 35–1000 °C. Of the material sample, 11 mg was placed in an alundum crucible and the weight (thermogravimetric, TG) and thermal (differential thermal, DT) curves were recorded during heating.

#### 2.2.6. Electrochemical Measurements

##### Samples activation

The MEA was placed in a standard measuring cell (FC-05-02, ElectroChem, Inc., Woburn, MA, USA.) with graphite current-collecting electrodes, an excess gas pressure of 0–2 atm., and an electronic resistance of less than 10 mΩ. Toray 060 standard carbon paper was used as a gas diffusion layer.

Before starting the main measurements, the MEA was activated as described in [55]. The potential of the electrode under study was not refined.

##### Measurements

Measurements were carried out under the conditions of a supply of moistened gases saturated with water at room temperature in a stoichiometric ratio (H_2_:O_2_ = 2) at atmospheric pressure. The temperature of the sample was changed in the range from room temperature to 120 °C. Electrochemical measurements were carried out using a two-electrode scheme with a closed anode (H_2_) and an opened cathode (O_2_) output. The current–voltage characteristics were recorded at a polarization rate of 10 mV/s.

The electrochemical tests were carried out in three versions:(1)open circuit voltage (OCV) vs. time;(2)voltammograms (VAC); and(3)the dependences of the current density in the potentiostatic mode at a voltage *E* = 0.65 V vs. time.

## 3. Results and Discussion

### 3.1. Proton Conductivity of the Compositional Membranes

The study of the proton conductivity of composite membranes of the Aquivion^®^-type (EW = 890 g-eq/mol) with DND Z+ was considered in detail in our previous work [39]. The optimum proton conductivity of composites with a DND Z+ content in the range of 0–5 wt.% DND is achieved at a low DND content of 0.5 wt.%, resulting in the value of 0.23 S/cm at 50 °C. Further increasing the DND content leads to a slight decrease in the proton conductivity of the composites, apparently due to a partial blocking of water-conducting channels, which is typical for many organic and inorganic fillers in the composition with perfluorinated sulfonic acid membranes [56].

The resulting parameters of the performed physical tests are summarized in Table 1.

### 3.2. Structural Studies by SANS

SANS was performed on the Aquivion^®^-type membranes (EW = 890 g-eq/mol) with DND Z+ in the range of 0–5 wt.% in air-dried conditions. Figure 2a shows the SANS curves dΣ/dΩ(*q*), where the ionomer peak at q ~2 nm^−1^ is detected for all studied samples. This peak corresponds to correlations of the neighboring conducting channels at fixed distances of ~3 nm, assembled in bundles, being structural elements of the membrane [52].

SANS on a membrane without DNDs was fitted using a model, (2), describing cylindrical channels in the polymer matrix and their correlations on several structural levels. Table 2 shows the resulting fitting parameters, and we focus on the most important structural parameters. The channels’ size (diameter) is 2$\sqrt{2}$*R_g_* = 0.7 nm, which is an inexact value, being outside the range of the SANS structural scale (1–100 nm) for the used *q*-range, but it agrees with our previous data for dry channels in short-side-chain membranes [52]. Therefore, the elementary channels are very thin, and they are curved and bent (*n* = 1.5). Correlations between dry neighboring channels in a single bundle are observed at a distance of *R*_2_ = 3.2 nm, corresponding to an ionomer peak on the SANS curve.

The position of the ionomer peak remains unchanged at the presence of DNDs (Figure 2a), which demonstrates that nanodiamonds generally do not disturb the basic channel structure of the membrane. DND particles are too large and do not enter inside the channels, while they are integrated into the membrane on larger-scale levels. The profile of the ionomer peak at the presence of the diamonds becomes smoother due to the additive contribution of the introduced nanodiamonds. The other reason may be that 4–5 nm DND particles could be associated on the outer surface of conducting channels, close to the SO_3_H groups of the copolymer, and may partially penetrate bundles of channels between neighboring channels, thus making correlations of channels less intensive.

The next structural level is described by R_4_ = 12.8 nm and corresponds to another prolonged peak in the q-range of 0.4–1.0 nm^−1^—a matrix knee (Figure 2b), which disappears at the presence of DNDs (Figure 2a). This wide peak is attributed as a degree of a crystallinity—an ordering of bundles of channels at the distance R_4_, which is typical for Aquivion^®^ in dry conditions in the range of ~10–20 nm. The disappearing of this peak may indicate that the membrane structure at these scales became more uniform at the presence of the DNDs. Likely, the DND particles can be incorporated between the bundles of conducting channels that breaks their ordering, but a more accurate interpretation seems to be the additively contributed intensity of the DNDs that greatly scatter in this q-range, if associated in clusters [57,58,59].

Both diamonds and polymers contribute to scattering, and an accurate separation of the contributions of these components from the measured data seems a challenging task. For a conditional representation of the distribution of diamond particles, we performed a subtraction of the curves. Figure 2c shows the differential SANS curves for the compositional membranes with DNDs obtained from the original data by the subtraction of the data for a pristine membrane without DNDs. It turned out that the distribution of DNDs in membranes obtained by drying DMF mixtures is similar to the association of diamond particles into branched chain clusters, which is observed in DND hydrosols [57,58,59]. At a low q, the differential curves conform the power law dΣ/dΩ(q) = *I*_0_·*q*^−*D*^ with a slope of *D* = 2.3, which corresponds to fractal objects—clusters of DND particles with a fractal dimension of 2.3. The area at *q* > 1 nm^−1^ also corresponds to a power law with a slope of 4.3, which characterizes primary diamond particles with sharp facets [58], which are well detected at a high concentration in the membrane. Apparently, both the membrane and diamond components, despite the electrostatic interaction in the DMF suspension, retain their entire structures upon drying, while integrating into each other. The inset in Figure 2c shows the forward scattering intensity *I*_0_ in the range of the smaller q, normalized to the DND concentration, as a function of the DND concentration in the membrane. This is an almost constant value, which decreases slightly after 0.5 wt.% DNDs, i.e., the additivity of the SANS intensity from a diamond component is somewhat disturbed. This slight decrease may correspond to an increase in the uniformity in the distribution of diamond particles on a scale of 10–100 nm, i.e., a decrease in the fraction of nanosized diamond clusters and the growth of submicron clusters, with an increase in the concentration of DNDs in membranes.

We can conclude that the forms of structuring typical to DND particles are compatible with the molecular (supramolecular) organization of the polymer matrix and diamonds are completely integrated into its structure, improving the mechanical properties and stability of the membranes, and at low concentrations, they enhance ionic conductivity.

### 3.3. Surface Structure of Membranes from Scanning Electron Microscopy

A series of membranes of the Aquivion^®^-type (EW = 890 g-eq/mol) with DND Z+ (0, 0.5, and 5 wt.% DND) was studied. On the surface of the membrane without diamonds, a cellular structure with globule sizes of approximately 150 nm is detected (Figure 3a), which is consistent with our atomic force microscopy (AFM) data [38]. On the surface of the composite membrane (0.5 wt.% DND), diamond clusters are found, mostly up to 200 nm in size and uniformly distributed on the membrane (Figure 3b). In the presence of 5 wt.% DND, diamonds on the membrane surface are in the form of clusters of up to 200–300 nm in size (Figure 3c,d). Thus, a 10-fold rise of DND content in the compositional membranes leads to a more dense distribution of DND clusters, resulting in the growth of the size of the submicron DND clusters, which confirms our assumption above, based on the SANS data. On the micron scale, diamonds are rather uniformly distributed. Moreover, they should be localized on the surface of hydrophobic polymer globules and in the spaces between globules in the hydrophilic phase of the copolymer or aggregated at the interface between the hydrophilic and hydrophobic phases. For comparison, according to the AFM data, a strongly inhomogeneous distribution of diamonds was observed on the compositional Aquivion^®^-type membrane with a negatively charged DND Z- when they were collected in large clusters at the boundary of the hydrophobic and hydrophilic phases [38]. At low concentrations (C_DND_ ≤ 1 wt.%) on a micron scale, the distribution of DNDs in the membrane is no longer regulated by the interactions of the diamond particles with each other (which leads to the formation of nanoscale chain clusters [57,58,59]), but rather, by the nature of the packing of the submicron polymer domains (diameter of ~150 nm), which form a regular cellular structure (SEM and AFM [38] data). In this case, the diamond nanoclusters predominantly fill the available gaps between the polymer domains, which is confirmed by small and very small angle neutron scattering data (SANS and VSANS) [38], which showed that the diamonds decorate polymer domains, modulating the contrast between the dense and loose areas of the polymer matrix. However, the fraction of the matrix volume available for filling with diamonds without disturbing the packing of the polymer domains is limited. Therefore, with an increase in the content of diamonds (C_DND_ > 1–3 wt.%, depending on the DND charge sign), their partial segregation occurs, and accumulations of submicron-scale diamonds are observed on the membrane surface. DND Z+ particles are more likely to associate with SO_3_^−^ groups of the hydrophilic phase of the copolymer, and so the distribution of such diamonds is more uniform. However, the polymer can only bind a limited number of diamonds through electrostatic attraction to its SO_3_^−^ groups. Unbound DND Z+ particles, similar to those observed in the case of Aquivion^®^ with the DND Z- sample, were probably adsorbed on the polymer, binding nonspecifically to the polymer in the form of large clusters while the DMF suspension was dried. An excess of diamonds thus hinders the proton conductivity. The optimal content of DND Z+ was 0.5 wt.%, above which a decrease in conductivity was observed due to the complication of the pathways for protons [39].

### 3.4. Mechanical Tests of Compositional Membranes

Aquivion^®^-type membranes (EW = 890 g-eq/mol) with DND Z+ in the range of 0–5 wt.% were studied. The measured parameters are shown in Figure 4, with the detailed analysis published in [39]. All tested membranes are low-modulus polymeric materials (a modulus of elasticity of ~200–300 MPa) with high-strain resources (ultimate deformation before destruction of 120–320%). The rigidity of the films (the modulus of elasticity and yield strength), as well as the ultimate tensile strength, demonstrate extremum at 0.5 wt.% of DND, being in a good agreement with the proton conductivity data.

The demonstrated effects for Aquivion^®^-type membranes with DND Z+ on the conductivity and mechanical parameters surpass those measured for Aquivion^®^ with DND Z- compositional membranes [38]. This fact confirms our assumption about a deeper integration of DND particles into a polymer matrix in the case of the positive charge of the DNDs.

### 3.5. Thermogravimetric Analysis (TGA)

The mechanism of thermal degradation of Aquivion^®^ and Nafion^®^ has been studied in a number of works [19,60,61]. In accordance with the results of these studies, the destruction of Aquivion^®^ occurs in several stages. The temperature limits in different works are somewhat different, which may be due to different heating rates and different brands of the studied samples. However, three main stages of Aquivion^®^ degradation in the H^+^ form have been identified. The total weight loss of all membranes up to 300 °C is associated with water evaporation (1). The weight loss in the range of 300–400 °C corresponds to the decomposition of the polymer side chains (2), while the weight loss above 400 °C is associated with the thermal oxidation of the backbone of the polymer membrane (3).

For the current study, samples of the membranes in an air-dried condition were used, meaning that they had equilibrium water content, which characterizes their water-retaining ability. The thermograms of the studied samples of Aquivion^®^-type composite membranes in an air-dried condition (Figure 5) in the low-temperature range demonstrate two areas of a thermal dehydration in the temperature range of 35–250 °C, expressed as steps on the TG and minima on the DTG curves, respectively. The low-temperature section 50–110 °C can be associated with the removal of free water and the interval 110–250 °C with the removal of bound water. Figure 5 demonstrates that the temperatures of the minima nearly coincide for the three studied samples and are ~72 °C and ~130 °C for the first and second peaks, respectively. This fact indicates the close values of their water-retaining capacities. The fractions of the removed water for the studied membranes are also close to each other. For the first peak, it is in the range of 3–4%; for the second peak, it is 3.5–4%. Both stages of dehydration are expectedly accompanied by endothermic effects (see the inset in Figure 5), which is consistent with the concept of water evaporation from the material.

The moisture content of the samples of different membranes has similar values. This indicates that the addition of DND Z+ does not significantly affect this parameter.

### 3.6. Electrochemical Tests

Figure 6, Figure 7 and Figure 8 demonstrate the characteristics obtained during the electrochemical tests for the three samples with different DND contents.

We note that the condition of the measured samples and, accordingly, their characteristics quite strongly depend on the operations performed earlier. In this regard, the measurements were carried out methodically, using the same algorithm of actions for all samples, when possible (dispersion, measurement sequence, etc.). Despite these measures, some scatter of the results was still observed, which is visible in the Figures. However, the overall picture allows us to trace certain patterns.

The dependences of the open circuit voltage on temperature for the studied samples are qualitatively similar: OCV tends to decrease with increasing the temperature. It is important to note that for the sample without DNDs (Figure 6a), the drop in OCV with the temperature is the strongest. At 100 °C, it has already dropped below 650 mV. The membrane with a 2.6 wt.% DND (Figure 8a) also shows a rather large drop in OCV with increasing the temperature, but it remains at ~770 mV at 120 °C. Finally, the sample with a 0.5 wt.% DND (Figure 7a) showed the greatest resistance of OCV to the temperature increase, and it did not fall below 880 mV. In our opinion, such dependences of OCV on the temperature characterize the water-retaining ability of the membranes. The greater the water-retaining ability, the smaller the drop in OCV with the temperature. The best water-retaining ability was demonstrated by the MEA with a membrane containing a 0.5 wt.% DND (Figure 7), which also correlates with the proton conductivity data and mechanical characteristics of such a composite.

When analyzing the VAC, it should be noted that a common qualitative pattern for all samples in the current–voltage characteristics is the nature of the change in the slope of the linear section with the increasing temperature. The linear section located in the middle of the VAC is usually attributed to the ohmic loss area [62]; it characterizes the proton resistance of the sample. Initially, as the temperature increases, the slope of the linear section of the VAC for all studied samples decreases, reflecting a decrease in the ionic resistance of the ionomer with the increasing temperature. With a further increase in temperature, a rather sharp increase in the slope occurs, apparently associated with a decrease in the humidification of the Aquivion^®^ ionomer. Quantitative estimates show that the sample without DNDs (Figure 6b) demonstrates a sharp increase in an ohmic loss starting at 80 °C. For the sample with a 2.6 wt.% DND (Figure 8b), a sharp increase in the ohmic loss begins only at 120 °C. The same behavior is demonstrated by the sample containing a 0.5 wt.% DND (Figure 7b). Thus, the analysis of the temperature dependence of the VAC shows that at 120 °C, the compositional samples containing the 0.5 and 2.6 wt.% DND in the membrane demonstrate the lowest ohmic loss associated with moisture loss compared to the sample without DNDs.

The dependences of the current density in the potentiostatic mode corresponding to the electrical efficiency of 0.5 (E = 0.65 V) are in good agreement with the temperature dependences of the OCV and VAC. The membrane without DNDs (Figure 6c) demonstrates a sharp drop in the current density to very low values (already at 80 °C), while two other samples showed a larger resistance to the elevated temperatures (Figure 7c and Figure 8c). Due to the drop in the OCV of the sample without DNDs (Figure 6a) to values of less than 0.65 V, the potentiostatic curves for the temperatures of 100 and 120 °C were taken only at E = 0.25 V.

The decrease in the OCV with the temperature is caused by two known reasons: an increase in hydrogen crossover across the membrane and a decrease in the moisture content of the ionomer. A decrease in the moisture content leads to the degradation of the interfacial area (Pt-ionomer-pores), the partially dropping out of the active surface of the electrode from the four-electrode electrochemical process, and because of this, the mixed potential is moved to lower values.

## 4. Conclusions

Compositional membranes of the Aquivion^®^-type with protonated detonation nanodiamonds (DND Z+), prepared by a solution casting technique, demonstrated excellent properties for use in hydrogen fuel cells in the temperature range of 22–120 °C. The optimal content of DNDs in the composite membranes was found based on physical and mechanical tests. The maximum rigidity of the films and the ultimate tensile strength at the stretching of composite membranes is achieved at a 0.5 wt.% of DNDs in the composite.

Small-angle neutron scattering of compositional membranes with DND Z+ has revealed the retaining of the elementary structure of the conducting channels in the polymer matrix. DND particles modify the polymer matrix, forming branched clusters of up to 200–300 nm in size, visualized by scanning electron microscopy, being uniformly distributed over the copolymer on the submicron scale.

The water-retaining capacity of the Aquivion^®^-type membrane at room temperature has not changed significantly in the presence of DNDs, according to the thermogravimetric analysis data.

The effect of detonation nanodiamonds, as fillers in the compositional Aquivion^®^-type membranes, on the membrane’s electrochemical characteristics at elevated temperatures is summarized as follows:the addition of DNDs had a significant effect on the electrochemical performance of the membrane at elevated temperatures;the compositional membrane with a 0.5 wt.% DND demonstrated the best water-retaining capacity;at 120 °C, the membranes with 0.5 and 2.6 wt.% DNDs showed the lowest ohmic loss due to moisture loss compared to the sample without DNDs; andthe sample without DNDs demonstrated a sharp drop in current density to very low values, already at 80 °C, while the compositional samples with DNDs showed greater resistance to elevated temperatures.

Thus, DND Z+ particles at the optimal content of a 0.5 wt.% significantly improved the functional properties and the performance of the compositional membranes of the Aquivion^®^-type. The obtained membranes are prospective for use in hydrogen fuel cells at temperatures of up to 120 °C.

## Figures and Tables

**Figure 1 membranes-12-00827-f001:**
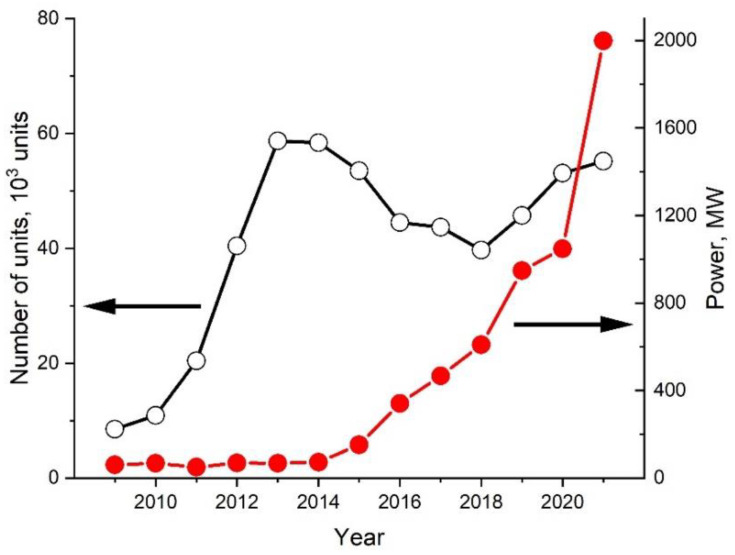
Proton exchange membrane fuel cells (PEMFC) device dynamics by year (number of units and power provided) (data adopted from [25]).

**Figure 2 membranes-12-00827-f002:**
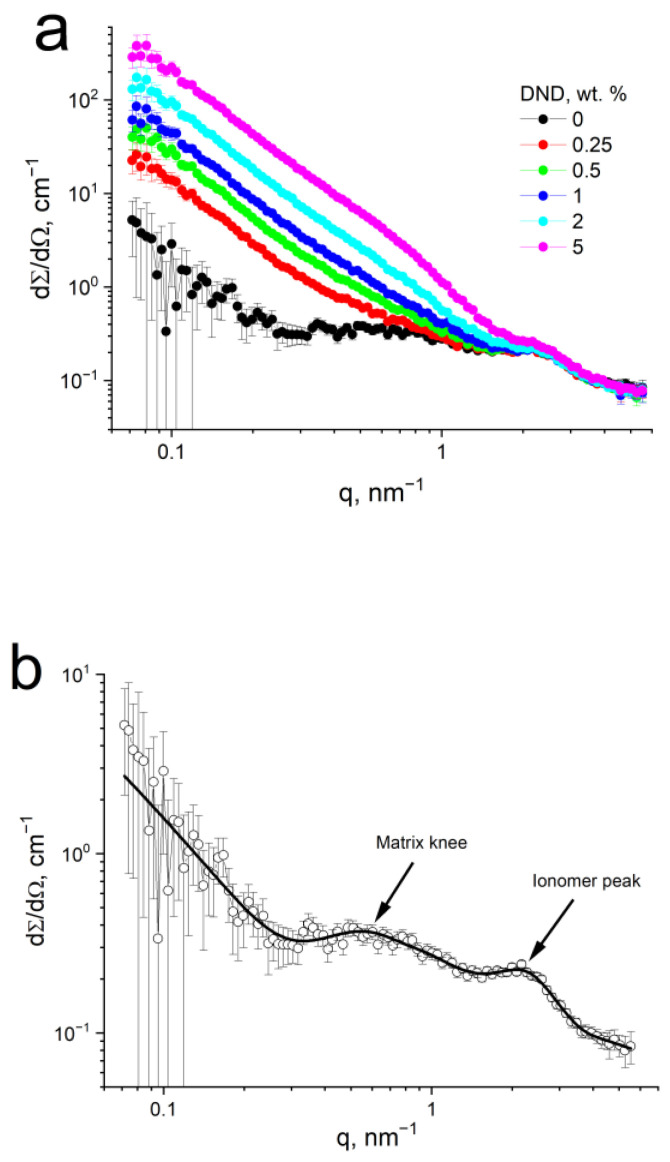
Small-angle neutron scattering (SANS) on Aquivion^®^-type membranes with detonation nanodiamonds (DND) Z+: (**a**) SANS curves of membranes with a DND content of 0–5 wt.%; (**b**) SANS on a sample without DNDs: the points are experimental data and the solid curve is a fitting result using Equation (2); and (**c**) differential SANS curves of membranes without DNDs subtracted from compositional membranes with DNDs, demonstrating the DND distribution in membranes (the points are experimental data, the solid lines are the power-law approximation, and the inset shows the intensity of the power-law approximation normalized by the DND content in membranes vs. the DND content).

**Figure 3 membranes-12-00827-f003:**
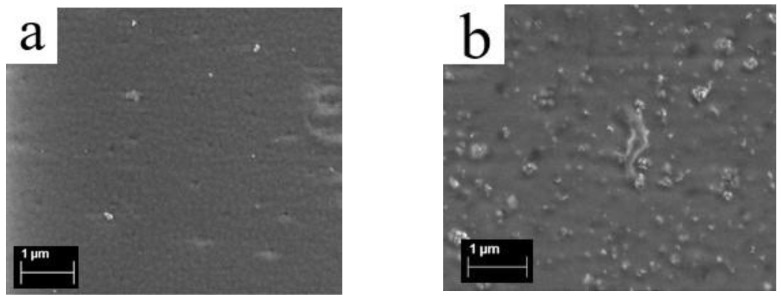
SEM images of the Aquivion^®^-type membranes (EW = 890 g-eq/mol) with DND Z+: (**a**) 0% DND (without nanodiamonds); (**b**) 0.5 wt.% DND; and (**c**,**d**) 5 wt.% DND (different scales).

**Figure 4 membranes-12-00827-f004:**
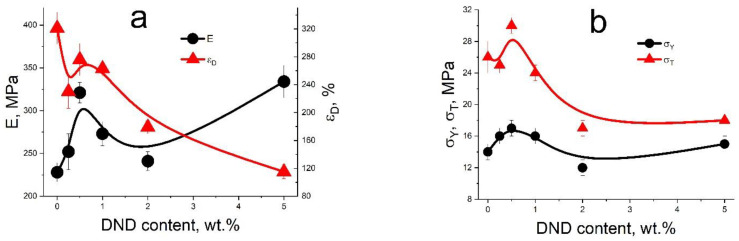
Mechanical properties of the compositional Aquivion^®^-type membranes (EW = 890 g-eq/mol): (**a**) Young’s modulus E and ultimate deformation before destruction ε_D_, and (**b**) yield strength σ_Y_ and ultimate tensile strength σ_T_.

**Figure 5 membranes-12-00827-f005:**
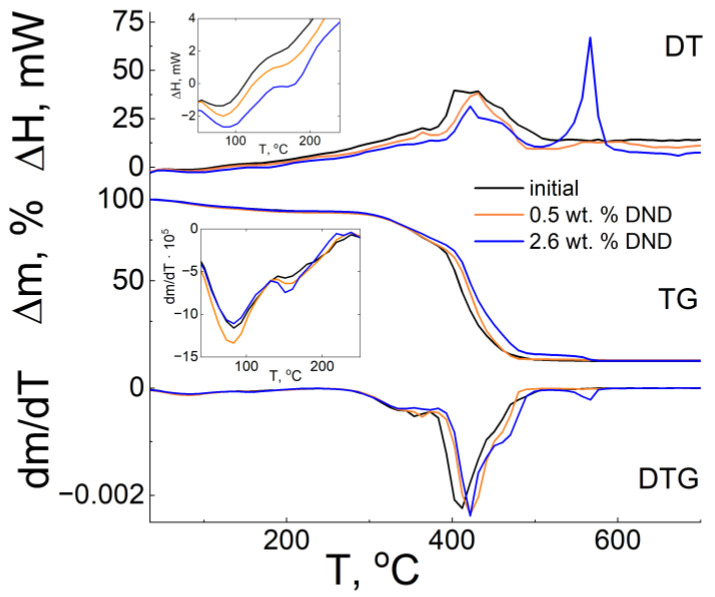
Differential thermal curves of the samples of various composite membranes of the Aquivion^®^-type (EW = 897 g-eq/mol) with DND Z+.

**Figure 6 membranes-12-00827-f006:**
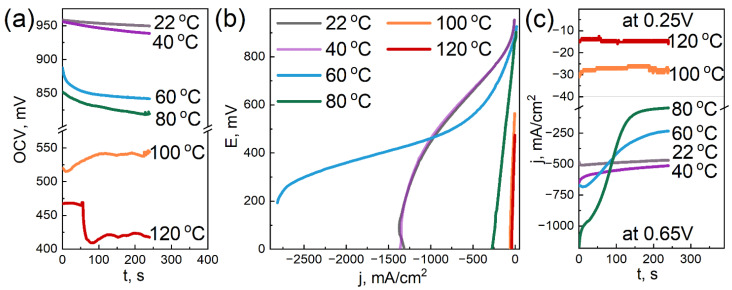
Electrochemical characteristics of the MEA with the Aquivion^®^-type membrane (EW = 897 g-eq/mol) without DNDs at different temperatures: open circuit voltage (OCV) vs. time (**a**); voltammograms (**b**); and current density in the potentiostatic mode at voltages of 0.65 and 0.25 V vs. time (**c**).

**Figure 7 membranes-12-00827-f007:**
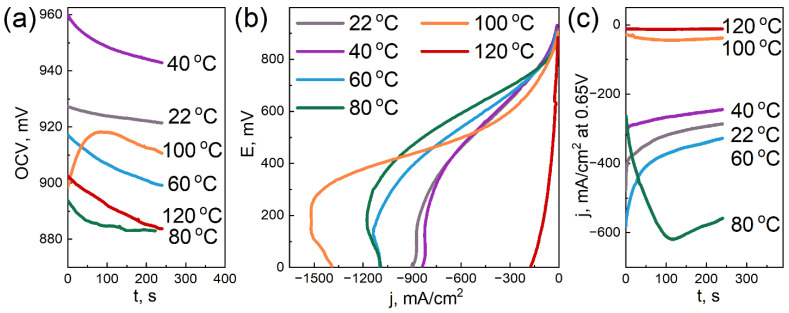
Electrochemical characteristics of the MEA with the Aquivion^®^-type membrane (EW = 897 g-eq/mol) with a 0.5 wt.% DND Z+ at different temperatures: OCV vs. time (**a**); voltammograms (**b**); and current density in the potentiostatic mode at a voltage of 0.65 V vs. time (**c**).

**Figure 8 membranes-12-00827-f008:**
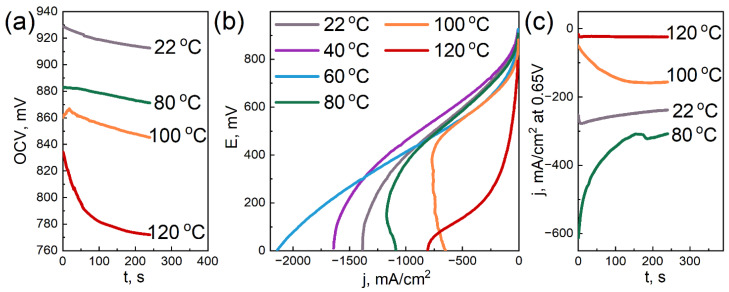
Electrochemical characteristics of the MEA with the Aquivion^®^-type membrane (EW = 897 g-eq/mol) with a 2.6 wt.% DND Z+ at different temperatures: OCV vs. time (**a**); voltammograms (**b**); and current density in the potentiostatic mode at a voltage of 0.65 V vs. time (**c**).

**Table 1 membranes-12-00827-t001:** Physical parameters for the Aquivion^®^-type membranes with and without detonation nanodiamonds (DND).

DND Content, wt.%	Water Uptake, wt.%	Ion Exchange Capacity, mmol/g	Proton Conductivity, S/cm	
			20 °C	50 °C
0	30.3	1.12	0.131	0.178
0.25	31.1	1.12	0.133	0.203
0.5	32.8	1.12	0.136	0.234
1.0	32.9	1.10	0.130	0.210
2.0	32.2	1.07	0.120	0.207
2.6	33.2	1.06	0.127	0.204
5.0	31.9	1.04	0.115	0.191

**Table 2 membranes-12-00827-t002:** Fitting parameters for Small-angle neutron scattering (SANS) on an Aquivion^®^-type membrane without detonation nanodiamonds (DND), using Equation (2).

Parameter	Value	Parameter	Value
*I*_0_, cm^−1^ nm^−^^3^	3.6 ± 1.0	*R*_2_, nm	3.2 ± 0.3
*n*	1.5 ± 0.4	*C* _3_	−0.6 ± 0.3
*R_g_*, nm	0.25	*R*_3_, nm	4.3 ± 0.4
*C* _1_	−1.2 ± 0.2	*C* _4_	0.10 ± 0.07
*R*_1_, nm	1.83 ± 0.15	*R*_4_, nm	12.8 ± 1.3
*C* _2_	0.8 ± 0.3	*B*, cm^−1^	0.068 ± 0.009

## Data Availability

Not applicable.

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
