# Peer review of "New Generation of Compositional Aquivion®-Type Membranes with Nanodiamonds for Hydrogen Fuel Cells: Design and Performance"

_membranes, 2022, doi:10.3390/membranes12090827_

Round 1
Reviewer 1 Report
The manuscript needs substantial changes to be accepted for Publication. Kindly consider the following
[1] The level of English language must be checked throughout the manuscript to make sure that the article is free from grammatical mistakes.
[2] “perfluorinated Aquivion® membranes with short side chains, which allow maintaining high conductivity at temperatures up to 130°C” – mention high “” proton conductivity”. Also, R is the protonic resistance of the membrane (Ohm),
[3] composites using functionalized nanofillers should improve the functional characteristics of membrane materials. – List out the integral functional characteristics of the membrane relative to the present study.
[4] The authors state that on the micron scale the diamond is more uniformly distributed, unlike the nano where there is cluster formation. More insight on the scientific principle with authentic reference is mandate
[5] Kindly refer to the following manuscript for more insight on the nano MEA and related work
Pethaiah, S.S., Kalaignan, G.P., Ulaganathan, M. and Arunkumar, J., 2011. Preparation of durable nanocatalyzed MEA for PEM fuel cell applications. Ionics, 17(4), pp.361-366.
The manuscript has numerous type errors, kindly make a through proof reading
Author Response
Dear reviewer,
We appreciate your detailed comments that helped us to improve the manuscript. We have revised the manuscript concerning your questions and here provide the responses. We have highlighted the changes in the manuscript according to your queries.
Best regards,
Yuri V. Kulvelis and co-authors
The manuscript needs substantial changes to be accepted for Publication. Kindly consider the following
[1] The level of English language must be checked throughout the manuscript to make sure that the article is free from grammatical mistakes.
Response. Thank you for the comment. We have checked the English grammar and made necessary changes.
[2] “perfluorinated Aquivion® membranes with short side chains, which allow maintaining high conductivity at temperatures up to 130°C” – mention high “” proton conductivity”. Also, R is the protonic resistance of the membrane (Ohm),
Response. Yes, you are right. We have added these clarifications.
[3] composites using functionalized nanofillers should improve the functional characteristics of membrane materials. – List out the integral functional characteristics of the membrane relative to the present study.
Response. This is a comment to the line 76. As we have shown, the introduction of 0.5% DND makes it possible to achieve a proton conductivity of the composite of 0.23 S/cm at 50°C, which is 28% higher than that of the original membrane without DND (line 289). Mechanical tests (Fig. 4) of the composites have shown that the introduction of 0.5% DND can significantly increase the Young's modulus and strength properties of membranes compared to an unfilled membrane. Thus, the introducing of the optimal amount of DND made it possible to increase the proton conductivity, water retention at elevated temperatures, and the physico-mechanical properties of composite membranes. The addition of DND has also a significant effect on the electrochemical characteristics of the membrane at elevated temperatures.
[4] The authors state that on the micron scale the diamond is more uniformly distributed, unlike the nano where there is cluster formation. More insight on the scientific principle with authentic reference is mandate
Response. The formation of nanoclusters is inherent to DND [refs. 57-59 and other numerous works]. But on a micron scale at low concentrations (CDND ≤ 1% wt.), the distribution of DND in the membrane is no longer regulated by the interactions of diamond particles with each other (which leads to the formation of nanoscale chain clusters [57-59]), but by the nature of the packing of submicron polymer domains (diameter ~ 150 nm), which form a regular cellular structure (SEM and AFM [ref. 38] data). In this case, diamond nano-clusters predominantly fill the available gaps between polymer domains, which is confirmed by small and very small angle neutron scattering data (SANS, VSANS) [38], which showed that diamonds decorate polymer domains, modulating the contrast between dense and loose areas of the polymer matrix. However, the fraction of the matrix volume available for filling with diamonds without disturbing the packing of polymer domains is limited. Therefore, with an increase in the content of diamonds (CDND > 1-3 wt. %, depending on the DND charge sign), their partial segregation occurs, and accumulations of submicron-scale diamonds are observed on the membrane surface. In the case of negatively charged DND they are more inclined to form micro-scale clusters in membranes [38] because such DND repulse from copolymer during the casting procedure from liquid dispersion. Positively charged DND particles, used in the current work, distribute more uniformly in the membrane due to their attraction to sulfonic groups of the copolymer. This is the reason of more uniform distribution of positive DND vs negative DND on a microscale. But on the nanoscale the cluster formation takes place, which do not allow DND particles to be located separately in the membrane, regardless their charge sign.
[5] Kindly refer to the following manuscript for more insight on the nano MEA and related work
Pethaiah, S.S., Kalaignan, G.P., Ulaganathan, M. and Arunkumar, J., 2011. Preparation of durable nanocatalyzed MEA for PEM fuel cell applications. Ionics, 17(4), pp.361-366.
Response. We have added a link at number 32. However, this article relates to the problems of depositing platinum nanocatalysts on the surface of Nafion membranes and is not directly related to the production of polymer membrane composites with. It is useful to compare the methods in the article with those used by us.
In our work, MEA were fabricated by applying a fine dispersion of components in an n-propanol-water mixture (Catalyst Inks) onto the surface of a proton-conducting membrane on both sides. To do this, the membrane was thermostated at 85 ± 5°C and the area of application of the electrode material 1x1 cm2 in size was limited by a stainless steel mask. The loading of components was controlled gravimetrically.
The authors [Pethaiah, S.S., Kalaignan, G.P., Ulaganathan, M. and Arunkumar, J., 2011. Preparation of durable nanocatalyzed MEA for PEM fuel cell applications. Ionics, 17(4), pp.361-366.] have modified and applied the impregnation-reduction method for depositing a platinum catalyst on a Nafion membrane. They showed that an electrode assembly with a platinum nanocatalyst (particle size 5.2 nm, concentration on the membrane surface 0.4 mg/cm2) obtained by a modified non-equilibrium impregnation-reduction method has better performance and durability than traditional assemblies. Next, an assembly was prepared by hot pressing from a platinum-coated Nafion membrane placed between gas diffusion electrodes with a 40% Pt/C catalyst applied.
In our work, we used the same loading of the membrane surface with the same catalyst (0.40±0.06 mg/cm2) as used by Pethaiah et al.
The aim of our work was not to improve the method of applying the catalyst to the membrane, but to improve the functional properties of the membrane by using the modifying additive (nanodiamonds). In our further studies, these goals and objectives are planned to be combined to achieve the best performance of fuel cells.
The manuscript has numerous type errors, kindly make a through proof reading
Thank you for the notice. We tried to fix the errors.

Reviewer 2 Report
The manuscript reported the novel composite PEMs based on perfluorinated Aquivion® copolymers modified by DND with positively charged surfaces for PEMFCs. It is found that with the increase in DND amount, a stabilizing effect of DND on the composite membranes is observed. The MEA based on the composite membranes working in the O2/H2 system demonstrated larger operational stability, lower proton resistance, and higher current densities at elevated temperatures compared to the pristine membranes without additives.
I consider the content of this manuscript will definitely meet the reading interests of the readers of the Membranes journal. However, the discussion and explanation should be further improved. Hence, I suggest giving a minor revision and the authors need to clarify some issues or supply some more experimental data to enrich the content. This could be comprehensive and meaningful work after revision.
1. For grammar issues, it is suggested that the author double-check the small grammar errors in the full text, especially the lack and redundant use of definite articles.
For example, Line 59, ‘PEMFCs are widely used in vehicles and stationary micro-combined heat and power (micro-CHP) plants’; Line 81, ‘DND, due to the chemical inertness of the core and the possibility of their surface functionalization, makes it possible to obtain particles with acidic…’; Line 95, ‘leads to growth of inhomogeneities distribution and conductivity lowering’; Line 111, ‘Based on the experience of our studies’; and so on. I cannot point out every error, that is why I suggest the authors double-check throughout the manuscript for possible language improvements.
2. For the Keywords, ‘Aquivion®’, ‘ion-conducting’, and ‘stabilizing effect’ should be added in order to attract a broader readership.
3. Line 40, ‘One of the most important areas in the decarbonization of the economics, along with solar, wind, and hydropower, is the development of hydrogen energy’. In fact, the development of hydrogen energy can be combined with renewable energy. Renewable energy sources such as solar energy and wind energy are unstable and intermittent during generation, and thus these valuable electric energies are difficult to apply continuously and stably. To tackle this issue, the employment of electrochemical energy generator systems is needed to improve the utilization rate and stability of renewable energy [ChemSusChem 15.1 (2022): e202101798]. Hence, hydrogen technology and fuel cells can be used combined with renewable energies.
4. Line 47-58, why does the perfluorosulfonic acid become the most widely used PEMs for PEMFC applications? This issue should be further clarified. PFSA membranes contain hydrophobic PTFE backbone and short side chains terminated with hydrophilic sulfonic acid (–SO3H) end groups. The –SO3H groups of Nafion may dissociate and deliver protons as charge carriers when the membrane is in the wet state, hence PFSA membranes possess high proton conductivity [Ionics 25.9 (2019): 4219-4229]. In addition, the rigid PTFE backbone provides the membrane with excellent chemical and mechanical stability.
5. Line 261, ‘11 mg of the material sample was placed in an alundum crucible, and the weight (thermogravimetric, TG) and thermal (differential thermal, DT) curves were recorded during heating.’ What is the original status of the materials sample? Is it dried before at high temperatures and stored in the inert atmosphere to avoid humidity influence? Or is it just dried in the airflow/normal air conditions? This information should be provided since it influences the final analysis of the TG results.
6. Line 408, ‘The total weight loss of all membranes up to 300°C is associated with water evaporation (1). The weight loss in the range of 300–400°C corresponds to the decomposition of the polymer side chains (2), while the weight loss above 400°C is associated with the thermal oxidation of the backbone of the polymer membrane.’
This conclusion is too hasty and inconsistent with the facts. My only question is, where is the sulfonic acid group? Doesn't the sulfonic acid group degrade? Or does the sulfonic acid group degrade with the side chain? Please note that the sulfonic group is not a carbon chain or organic group. The first mass loss (1), observed at 30 < T < 150 °C, is associated with the loss of residual water from the membranes may be correct. While the second thermal event (2), observed in the region between ca. 150 and 250 °C should be associated with the loss of the -SO3H groups [Solid State Ionics 319 (2018): 110-116]. I suggest using the IEC to calculate how the weight loss of -SO3H in Aquivion should be and use the value to locate the accurate temperature range.
7. For the electrochemical measurement, what is the flow rate of the gas, and what is the backpressure? Why no polarization curve result is shown [see Figure 10, Ionics 27.4 (2021): 1653-1666, and Figure 8 of International Journal of Hydrogen Energy 45.8 (2020): 5526-5534]? How about the maximum current density, power density, and high-frequency resistance of the tested MEAs?
8. How about the oxidative stability of the prepared membranes, for example, in Fenton reagents? In addition, the ion-exchange capacity, water uptake, and swelling ratio of the prepared membranes should be summarized in a table, including all the essential physical-chemical properties.
Author Response
Dear reviewer,
We appreciate your detailed comments that helped us to improve the manuscript. We have revised the manuscript concerning your questions and here provide the responses. We have highlighted the changes in the manuscript according to your queries.
Best regards,
Yuri V. Kulvelis and co-authors
The manuscript reported the novel composite PEMs based on perfluorinated Aquivion® copolymers modified by DND with positively charged surfaces for PEMFCs. It is found that with the increase in DND amount, a stabilizing effect of DND on the composite membranes is observed. The MEA based on the composite membranes working in the O2/H2 system demonstrated larger operational stability, lower proton resistance, and higher current densities at elevated temperatures compared to the pristine membranes without additives.
I consider the content of this manuscript will definitely meet the reading interests of the readers of the Membranes journal. However, the discussion and explanation should be further improved. Hence, I suggest giving a minor revision and the authors need to clarify some issues or supply some more experimental data to enrich the content. This could be comprehensive and meaningful work after revision.
- For grammar issues, it is suggested that the author double-check the small grammar errors in the full text, especially the lack and redundant use of definite articles.
For example, Line 59, ‘PEMFCs are widely used in vehicles and stationary micro-combined heat and power (micro-CHP) plants’; Line 81, ‘DND, due to the chemical inertness of the core and the possibility of their surface functionalization, makes it possible to obtain particles with acidic…’; Line 95, ‘leads to growth of inhomogeneities distribution and conductivity lowering’; Line 111, ‘Based on the experience of our studies’; and so on. I cannot point out every error, that is why I suggest the authors double-check throughout the manuscript for possible language improvements.
Response. Thank you for pointing out the errors. We have checked the text and tried to make it better.
- For the Keywords, ‘Aquivion®’, ‘ion-conducting’, and ‘stabilizing effect’ should be added in order to attract a broader readership.
Response. Thank you for suggestion. We have added these keywords.
- Line 40, ‘One of the most important areas in the decarbonization of theeconomics, along with solar, wind, and hydropower, is the development of hydrogen energy’. In fact, the development of hydrogen energy can be combined with renewable energy. Renewable energy sources such as solar energy and wind energy are unstable and intermittent during generation, and thus these valuable electric energies are difficult to apply continuously and stably. To tackle this issue, the employment of electrochemical energy generator systems is needed to improve the utilization rate and stability of renewable energy [ChemSusChem 15.1 (2022): e202101798]. Hence, hydrogen technology and fuel cells can be used combined with renewable energies.
Response. Thank you for the comment. We have added a few words in the introduction as follows:
At the same time, the development of hydrogen energy can be combined with the use of renewable energy. Hydrogen technologies and fuel cells can be effectively used in combination with renewable energy sources (solar, wind energy) to increase the stability of energy generation using renewable sources [10].
- Line 47-58, why does the perfluorosulfonic acid become the most widely used PEMs for PEMFC applications? This issue should be further clarified. PFSA membranes contain hydrophobic PTFE backbone and short side chains terminated with hydrophilic sulfonic acid (–SO3H) end groups. The –SO3H groups of Nafion may dissociate and deliver protons as charge carriers when the membrane is in the wet state, hence PFSA membranes possess high proton conductivity [Ionics 25.9 (2019): 4219-4229]. In addition, the rigid PTFE backbone provides the membrane with excellent chemical and mechanical stability.
Response. Again, thank you for the comment. This is a reasonable question which arises in modern works not often.
Due to the perfluorinated chemical structure of the PFSA membrane backbone, which is stable against peroxide attack, a high operational durability of such membranes and composites based on them during the operation in a hydrogen fuel cell is achieved, reaching tens of thousands of hours [5] and [Danilczuck M., Lancucki L., Schlick S., Hamrock S.J., Haugen G.M. In-depth profiling of degradationin processes in a fuel cell: 2D spectral-spatial FTIR spectra of Nafion membranes. ASC Macro Lett. 2012, 1, 280-285. DOI: 10.1021/mz200100s].
- Line 261, ‘11 mg of the material sample was placed in an alundum crucible, and the weight (thermogravimetric, TG) and thermal (differential thermal, DT) curves were recorded during heating.’ What is the original status of the materials sample? Is it dried before at high temperatures and stored in the inert atmosphere to avoid humidity influence? Or is it just dried in the airflow/normal air conditions? This information should be provided since it influences the final analysis of the TG results.
Response: The original samples were in an air-dry condition (dried in normal air conditions), as it is mentioned in the Results and discussion (section 3.5. “Thermogravimetric analysis (TGA)”). We have also added this to the section 2.2.5.
Added this information to paragraph 2.2.5. thermogravimetric analysis. Relative humidity is about 60%.
- Line 408, ‘The total weight loss of all membranes up to 300°C is associated with water evaporation (1).The weight loss in the range of 300–400°C corresponds to the decomposition of the polymer side chains (2), while the weight loss above 400°C is associated with the thermal oxidation of the backbone of the polymer membrane.’
This conclusion is too hasty and inconsistent with the facts. My only question is, where is the sulfonic acid group? Doesn't the sulfonic acid group degrade? Or does the sulfonic acid group degrade with the side chain? Please note that the sulfonic group is not a carbon chain or organic group. The first mass loss (1), observed at 30 < T < 150 °C, is associated with the loss of residual water from the membranes may be correct. While the second thermal event (2), observed in the region between ca. 150 and 250 °C should be associated with the loss of the -SO3H groups [Solid State Ionics 319 (2018): 110-116]. I suggest using the IEC to calculate how the weight loss of -SO3H in Aquivion should be and use the value to locate the accurate temperature range.
Response. Thank you for the valuable comment. This type of membranes is known to contain two types of water molecules – free and bound water. Bound water evaporates at higher temperatures than free water. Basing on the literature data, we can say, that the elimination of sulfonic acid groups occurs at higher temperatures after the evaporation of water. Our derivatograms show a weight loss effect that starts at about 261°C, which can be attributed to the loss of sulfonic acid groups.
We can make some estimations. According to EW of our copolymer (EW = 897 ± 3 g-eq/mol), the weight loss of sulfonic acid groups should be about 9 %. In the second temperature range of 110–250°C, the weight loss was 3.5–4%. The next peak (Fig. 5, DTG) for different membranes has a different interval: ~261-343 °C for the original membrane without DND; ~ 261-352 °C for membrane with 0.5% DND, ~ 261-354 °C for membrane with 2.6% DNA. The weight loss, corresponding to this last peak in all three cases is ~ 11%, which is close to the content of the sulfonic acid groups in the material.
- For the electrochemical measurement, what is the flow rate of the gas, and what is the backpressure? Why no polarization curve result is shown[see Figure 10, Ionics 27.4 (2021): 1653-1666, and Figure 8 of International Journal of Hydrogen Energy 45.8 (2020): 5526-5534]? How about the maximum current density, power density, and high-frequency resistance of the tested MEAs?
Response. The hydrogen flow rate was 15 mL/min. The backpressure was 2-3 mmH2O.
Polarization curves (voltammograms), are shown at Fig. 6-8 b. The maximum current density (seen from voltammograms) was different for various samples at different temperatures and reached ~ 2800 mA/cm2 for membrane without DND at 60°C. Power density can be found from voltammograms, but since the aim of the work was to study the moisture resistance of membranes at elevated temperatures, and not the catalyst, power characteristics were not additionally defined.
The high-frequency resistance was ~ 0.1-0.15 Ohm.
- How about the oxidative stability of the prepared membranes, for example, in Fenton reagents? In addition, the ion-exchange capacity, water uptake, and swelling ratio of the prepared membranes should be summarized in a table, including all the essential physical-chemical properties.
Response. We have performed Fenton test only for pure Aquivion®-type membrane, without DND. The oxidative stability of the prepared membrane was confirmed. The membrane was placed in 100 cm3 of a 10% H2O2 solution with the addition of 0.0006 g of FeSO4 at 90°C for 2 hours. At the end of the experiment, the samples were washed with distilled water to a constant value of pH 6–7. A visual inspection of the membrane was carried out and the specific volumetric electrical resistance was measured. No defects were visually observed, and the proton conductivity of the membrane before and after the test was within the experimental error (5-7%). We have not yet performed Fenton tests for compositional membranes. Thus, we do not mention this result in the manuscript.
The main properties are summarized in the table, which is added to the manuscript (Table 1).
|
DND content, wt. % |
Water uptake, wt. % |
IEC, mmol/g |
Proton conductivity, S/cm |
|
|
20°Ð¡ |
50°Ð¡ |
|||
|
0 |
30.3 |
1.12 |
0.131 |
0.178 |
|
0.25 |
31.1 |
1.12 |
0.133 |
0.203 |
|
0.5 |
32.8 |
1.12 |
0.136 |
0.234 |
|
1.0 |
32.9 |
1.10 |
0.130 |
0.210 |
|
2.0 |
32.2 |
1.07 |
0.120 |
0.207 |
|
2.6 |
33.2 |
1.06 |
0.127 |
0.204 |
|
5.0 |
31.9 |
1.04 |
0.115 |
0.191 |

Round 2
Reviewer 1 Report
The manuscript needs few more clarification
1-In 365 the authors state " This observation was further confirmed by SEM method." kindly provide more clarity
2-Table-2 kindly check the unit of Io
3-Make a comprehensive proof-reading throughout the section, especially the conclusion part can be well organized
Author Response
Dear reviewer,
Thank you for the comments that helped us to improve the manuscript. Here we provide the responses and the revised version of the manuscript.
Best regards,
Yuri V. Kulvelis and co-authors
1-In 365 the authors state " This observation was further confirmed by SEM method." kindly provide more clarity
Response. Thank you for the comment. Our statement mean that it confirms SANS results stated above: “…decrease in the fraction of nanosized diamond clusters and the growth of submicron clusters, with an increase in the concentration of DND in membranes”.
We have deleted this sentence “This observation was further confirmed by SEM method.” from the SANS section 3.2 and added the following to the SEM section 3.3: “Thus, 10-fold rise of DND content in the compositional membranes leads to more dense distribution of DND clusters, resulting in the growth of the size of submicron DND clusters, which confirms our assumption above, based on the SANS data.”
2-Table-2 kindly check the unit of Io
Response. The unit is right (cm-1nm-3). We are careful in using the correct units. In the model used (eq. 2) the left part, a scattering cross-section dS/dW(q) has the dimension of cm-1. Ri and Rg are expressed in nm, q – nm-1. So, qRi and qRg are non-dimensional, and the structural factor (a sum in large brackets in eq. 2) and the expression under the exponent are non-dimensional. Thus, I0Rg3 has a dimension of cm-1, wherefrom I0 has a unit of cm-1nm-3. It is possible to transform this to cm-4 or nm-4, but it does not make sense, since a scattering cross-section is traditionally expressed in cm-1 and a momentum transfer q – in nm-1 (in our case) or in Å-1. The physical meaning of I0 (forward scattering or a scattering cross-section to zero angle) is a cross-section (intensity) of scattering per unit volume, so leaving cm-1nm-3 is reasonable. It characterizes the scattering ability of structural elements. We have added the meaning of I0 to the experimental part.
3-Make a comprehensive proof-reading throughout the section, especially the conclusion part can be well organized
Response. We have made some changes in the manuscript and tried to improve the conclusion. Thank you again for the review. We appreciate your valuable comments.
